# Accuracy and Safety of Ultrasound-Guided Core Needle Biopsy of Soft Tissue Tumors in an Outpatient Setting: A Sarcoma Center Analysis of 392 Consecutive Patients

**DOI:** 10.3390/cancers13225659

**Published:** 2021-11-12

**Authors:** Helene Weigl, Peter Hohenberger, Alexander Marx, Nikolaos Vassos, Jens Jakob, Christian Galata

**Affiliations:** 1Department of Surgery, Universitätsmedizin Mannheim, Medical Faculty Mannheim, Heidelberg University, 68167 Mannheim, Germany; helene.weigl@umm.de (H.W.); nikolaos.vassos@umm.de (N.V.); jens.jakob@med.uni-goettingen.de (J.J.); christian.galata@unimedizin-mainz.de (C.G.); 2Division of Surgical Oncology & Thoracic Surgery, Department of Surgery, Universitätsmedizin Mannheim, Medical Faculty Mannheim, Heidelberg University, 68167 Mannheim, Germany; 3Institute of Pathology, Universitätsmedizin Mannheim, Medical Faculty Mannheim, Heidelberg University, 68167 Mannheim, Germany; alexander.marx@umm.de; 4Department of General, Visceral and Pediatric Surgery, University Medical Center Göttingen, 37073 Göttingen, Germany; 5Division of Thoracic Surgery, Academic Thoracic Center Mainz, University Medical Center Mainz, Johannes Gutenberg University Mainz, 55131 Mainz, Germany

**Keywords:** core needle biopsy, soft tissue sarcoma, accuracy, safety

## Abstract

**Simple Summary:**

In patients with unclear soft tissue lesions, treatment planning largely depends on histology. Core needle biopsy is the diagnostic standard in these cases. The aim of this retrospective study was to investigate accuracy and safety of ultrasound guided core needle biopsy at a tertiary referral center. We show that ultrasound guided biopsy was feasible as a one stop shop procedure in an outpatient setting in 87.6% of the cases. The rate of conclusive biopsies was 88.5%. After surgical resection, the dignity, tumor type and histopathological grading of the biopsy matched one of the resection specimens in 97.2%, 92.7% and 92.5% of the cases, respectively. Major complications occurred in 0.8% of the cases. With this study, we confirm that ultrasound guided core needle biopsy is safe, effective and can be performed at the first outpatient presentation to speed up the diagnostic and therapeutic cascade in sarcoma patients.

**Abstract:**

Background: The aim of this study was to investigate diagnostic accuracy, safety and histologic results of ultrasound guided core needle biopsy (CNB) in patients with soft tissue lesions (STL) at a tertiary referral center. Methods: A retrospective analysis of all consecutive patients undergoing ultrasound guided CNB for STL at our sarcoma outpatient service between January 2015 and August 2020 was performed. Results: A total of 392 patients were identified. Main histologic entities were sarcomas, lipomas and desmoid tumors. Biopsy was performed in an outpatient setting in 87.6% of the cases. Conclusive biopsies were obtained in 88.5% of the cases. In patients who underwent surgical resection after CNB, the concordance of dignity, tumor entity and histopathological grading between biopsy and resection specimen were 97.2%, 92.7% and 92.5% respectively. The risk of inconclusive CNB was highest in intraabdominal or retroperitoneal tumors (19.5%) and lowest in lesions at the lower extremity (4.4%). Major complications after CNB occurred in three cases (0.8%). No case of biopsy tract seeding was observed during the study period. Conclusions: Ultrasound guided CNB for STL at first presentation in a dedicated surgical outpatient setting is a safe procedure and yields a high diagnostic accuracy.

## 1. Introduction

In soft tissue lesions (STL), treatment planning largely depends on tumor entity. This is particularly true for patients with suspected soft tissue sarcomas where histotype and grading are essential for adequate multimodal treatment. In unclear STL, there are numerous differential diagnoses, each requiring different therapeutic strategies. For this reason, histologic confirmation is critical in STL before initiating treatment. Core needle biopsy (CNB) is recommended for patients with suspected soft tissue sarcoma in retroperitoneal locations as well as at the trunk and the extremities [1,2]. Alternatively, incisional biopsy may be considered in certain patients, depending on the experience of the treating center. A number of studies have analyzed the role of different biopsy techniques in bone tumors and STL before [3,4,5,6,7,8,9,10,11,12,13,14]. A recent systematic review and meta-analysis showed that CNB in soft tissue sarcoma has a high diagnostic accuracy and is superior to incisional biopsy in regard to complications [15]. The risk of biopsy tract seeding is extremely low with this procedure [16]. While CNB is an accepted standard to obtain biopsies from solid tumors, the approaches used in clinical practice vary considerably and range from freehand CNB to contrast enhanced and non-enhanced ultrasound guided CNB to CT guided CNB. The aim of this study was to evaluate diagnostic accuracy and complications of ultrasound guided CNB in a large cohort of patients with STL at a tertiary referral center with the ability to perform tumor biopsy during the first outpatient contact. The main results were that ultrasound guided CNB is safe, effective and can be performed in unselected patients at the first outpatient presentation with a high diagnostic accuracy.

## 2. Materials and Methods

### 2.1. Patient Data

The Sarcoma Center at the Mannheim Cancer Center (MCC) is a certified tertiary referral center for soft tissue sarcomas, gastrointestinal stromal tumors (GIST), desmoid tumors and bone sarcomas. About 150 patients are treated surgically each year. All patients who underwent ultrasound guided percutaneous CNB for STL between January 2015 and August 2020 were eligible for this study. To identify potential patients, a retrospective analysis of all medical records of the outpatient clinic for soft tissue tumors was performed. Furthermore, the records of surgical procedures were searched both manually and by electronic database query using the German modification (OPS) of the International Classification of Procedures in Medicine (ICPM) to identify patients who underwent CNB (OPS codes 1-502.0 to 1-502.9). Exclusion criteria for ultrasound guided CNB were non-identifiability of the target structure on ultrasound or a close location of the target structure to blood vessels or hollow organs. Anticoagulation with acetylsalicylic acid was not a contraindication for ultrasound guided CNB; in patients with therapeutic anticoagulation, therapy was usually paused before biopsy (24–48 h). Agents used for anticoagulation were acetylsalicylic acid, direct oral anticoagulants (Rivaroxaban, Apixaban, Edoxaban), Phenprocoumon or low molecular weight heparin in therapeutic dosage (Table 1). Patients who received medical thrombosis prophylaxis alone were not regarded as patients on anticoagulation. Histopathological examination of all specimens was routinely performed by dedicated pathologists according to the World Health Organization (WHO) and Fédération Nationale des Centres de Lutte Contre le Cancer (FNCLCC) classification systems.

### 2.2. Ultrasound Guided CNB

Ultrasound guided CNB was conducted under local anesthesia using sterile conditions according to standard operating procedures. In brief, the lesions intended for biopsy were visualized using a 2D curved (4 MHz) or linear (8.5 MHz) ultrasound transducer. Biopsies were taken under direct ultrasound control with a reusable core biopsy instrument (Bard Magnum Biopsy System, Becton, Dickinson and Company, Franklin Lakes, NJ, USA) using a 12G × 100 mm or 18G × 200 mm core needle and penetration depths of 15 mm or 22 mm, depending on lesion size and location. Typically, pre-interventional magnetic resonance imaging (MRI) and/or computed tomography (CT) scans with contrast media were available. In larger and heterogeneous tumors, we selected the site of biopsy according to the area of best contrast uptake. At least 2 to 3 core samples of tissue with an aspired length of 1.5 to 1.9 cm were obtained. After the CNB, patients were monitored for at least 20 min, followed by clinical examination and sonographic control of the puncture site. Major complications from CNB were defined as complications requiring hospital admission or interventional or operative management. A conclusive CNB was defined as a biopsy that allowed histopathological classification of the lesion and/or determination of further clinical management of the patient, whereas a CNB was defined as inconclusive if histopathological classification of the lesion was not possible or a re-biopsy was necessary to determine further clinical management. Dignity between biopsy and resection specimen was considered concordant if the diagnostic category (benign or malignant) was identical in both cases. Tumor entity was considered concordant between biopsy and resection specimen when the identical histopathologic tumor type was diagnosed in both cases by the pathologist. As a standard, all CNB were performed by or under supervision of board-certified surgeons, and all CNB operators had undergone appropriate ultrasound training.

### 2.3. Statistical Analysis

The mean and standard deviation were calculated for quantitative variables. The median together with the interquartile range (IQR) were presented for skewed or ordinally scaled parameters. Qualitative variables were quoted as absolute numbers and relative frequencies. The Student’s t test, the Mann–Whitney U test and the χ2 test or Fisher’s exact test were used, as appropriate. All statistical tests for the comparison of two groups were two-tailed. A test result was considered statistically significant if *p* < 0.05. A receiver operator characteristic (ROC) analysis was performed to determine the optimal cut-off value for the variable lesion size. Statistical analyses were performed using IBM SPSS Statistics (version 25, IBM Corp., Armonk, NY, USA).

## 3. Results

### 3.1. Patients’ Characteristics

A total of 392 consecutive patients fulfilled the criteria described above. Clinical characteristics of the patients are shown in Table 1. The median age of patients was 59 (46–71) years, the gender ratio was balanced. In 23.7% of the cases, a malignant diagnosis was known prior to biopsy and CNB was performed to confirm local or distant recurrence or to obtain additional tissue for molecular studies in the course of a multimodal therapy. In 76.3% of the cases, biopsy was performed to histologically verify a new lesion in patients without a history of malignancy. Patients with a history of malignancy were significantly older than patients without malignant disease (61 ± 17.1 years vs. 55.8 ± 16.9 years, *p* = 0.009) and underwent CNB for significantly smaller lesions (7.7 ± 6.4 cm vs. 9.9 ± 6.5 cm, *p* = 0.005).

Overall, the median time from diagnosis of the STL to CNB was 2 (1–5) months. Time to biopsy in patients with known malignancy was shorter than in patients without malignancy (7.9 ± 23.8 vs. 14.9 ± 49.8 months, *p* = 0.216). In patients where malignancy was confirmed after CNB, the time between diagnosis and CNB was significantly shorter than in patients with non-malignant histology (7.1 ± 22.5 vs. 21.7 ± 64.7 months, *p* = 0.011). In cases where malignancy was diagnosed after CNB, patients were significantly older than patients with non-malignant histology (64.5 ± 15.7 vs. 50.7 ± 15.8 years, *p* < 0.001) and had significantly larger lesions (11.0 ± 7.2 vs. 7.9 ± 5.2 cm, *p* < 0.001). Most patients in this study underwent CNB for lesions of the trunk or in intraabdominal and retroperitoneal locations (*n* = 223, 56.8%), whereas lesions of the extremities accounted for 43.1% of the cases (*n* = 169).

### 3.2. Outpatient and Inpatient Setting

CNB was performed in an outpatient setting in 87.6% of the cases. Only 12.5% of the cases underwent biopsy as inpatients. Of the 12.5% of patients who underwent biopsy as inpatients, only approximately one third (36.7%) were admitted to the hospital for CNB and two-thirds (63.3%) were hospitalized for reasons other than CNB, and biopsy was performed during the same hospital stay for patient convenience. In 15.8% of the patients, anticoagulant therapy was in place at the time of CNB. More than three quarters of CNB (*n* = 292, 75.8%) were performed 11 physicians who carried out ≥10 CNB during the observation period. CNB operators with less experience performed the procedure under supervision of an experienced CNB operator. The number of biopsies stratified by CNB operator experience is shown in Figure A1.

### 3.3. Complications

Major complications after CNB were observed in three cases (0.8%). One patient was diagnosed with a colonic perforation after CNB of an intraabdominal lesion and underwent exploratory laparotomy for peritonitis. In another case, CNB of a mass on the thoracic wall resulted in a pneumothorax, which required insertion of a thoracic drainage. In the third patient, a small bowel perforation was suspected after CNB of an intraabdominal mass. This patient was managed non-operatively; however, hospital admission and intravenous administration of antibiotics and analgesics was required. All major complications were observed in lesions on the trunk or in abdominal or retroperitoneal locations; no major complication was observed after CNB on the extremities. None of the patients with severe complications were on anticoagulation at the time of biopsy, and none of the patients who underwent CNB under anticoagulation (15.8%) experienced a serious bleeding complication. A biopsy was considered as performed under anticoagulation if the corresponding drug was not paused timely before the intervention. No patient death from CNB was recorded. No case of biopsy tract seeding was observed.

### 3.4. Diagnostic Accuracy and Histopathology

Conclusive histological diagnosis after CNB was obtained in 88.5% of the cases (*n* = 347), whereas 11.5% of CNB resulted in inconclusive histological result (Figure 1a). In patients with inconclusive histology, necrosis was given as the reason in five cases (11.1%) in the pathology report and hemorrhage in zero cases. Of conclusive CNBs, 52.7% (*n* = 183) were classified as malignant disease, while 47.3% (*n* = 164) were non-malignant (Figure 1b). In patients with malignant solid tumors, CNB was performed on the primary tumor in 68.1% of cases, whereas in 31.9% of cases it was a biopsy of a local recurrence or distant metastasis (Figure 1c). Histologic entities of conclusive CNBs are shown in Figure 2a. The most common malignant tumor types were sarcomas (37.5%), carcinomas (5.5%), lymphomas (4.9%) and GIST (3.2%). The most frequent non-malignant lesions were lipomas (17.6%) and desmoid tumors (8.1%). In the sarcoma group, more than half (60%) of the histologic subtypes were liposarcomas (40.8%) and pleomorphic sarcomas (19.2%) (Figure 2b). For 107 sarcomas diagnosed by CNB, FNCLCC grading was available. Of those, 70% were classified as high-grade (grades 2 and 3, Figure 2c).

### 3.5. Factors Associated with Inconclusive CNB

The rate of inconclusive CNB varied significantly between lesion locations (*p* = 0.006). The highest rate of inconclusive CNBs was observed in tumors in intraabdominal or retroperitoneal locations (19.5%, *n* = 15), while the rate of inconclusive CNB was 13.7% (*n* = 20) at the trunk, 12.5% (*n* = 4) at the upper extremity and lowest at the lower extremity with 4.4% (*n* = 6). Clinical parameters that might be associated with the occurrence of a non-conclusive biopsy were therefore examined separately for lesion location (Table 2). The only factor significantly associated with inconclusive biopsy was the size of the lesion, which was the case for tumors of the lower extremity (*p* = 0.036) as well as for tumors at the trunk (*p* = 0.006). A ROC analysis was performed aiming to determine an optimal cut off value for the lesion size that separates patients with conclusive CNB from patients with non-conclusive CNB. The highest Youden’s J was observed for a lesion size < 6.25 cm with a sensitivity of 84.6% and specificity of 57.4%. The corresponding ROC curve is shown in Figure 3.

### 3.6. CNB vs. Final Histopathology after Surgical Resection

In 179 patients with conclusive CNB, the histopathological diagnosis obtained by CNB could be compared to the final histopathology after surgical resection of the STL. In 174 of these cases (97.2%), the dignity of the lesion was concordant between CNB and final histopathology after surgery. Concordance of tumor entity between biopsy and final histopathology was observed in 92.7% of the cases (*n* = 166). In 80 patients, comparison of FNCLCC grading between CNB and resection specimen could be conducted. The grading was identical in 74 cases (92.5%). In the six patients where a different grading was observed, the grading on the surgical specimen was higher in five patients; in one case a lower grading was reported after resection.

## 4. Discussion

We present data on 392 consecutive patients who underwent ultrasound guided CNB of STL during a 6-year period. This is one of the largest single center analyses on the subject. Other studies on CNB commonly report less than 200 patients and often include different technical approaches (freehand, ultrasound guided, CT guided) or include patients with both bone lesions and STL [17,18,19,20,21,22,23,24]. The study by Strauss et al. reported 530 patients, but all with palpable masses that underwent freehand CNB without image guidance [25]. Hoeber et al. reported 570 cases of limb and limb girdle soft tissue tumors, including 523 Tru-cut biopsies, but without commenting on the use of imaging [26]. According to national and international guidelines, imaging should be carried out prior to biopsy to select the proper location.

Our data show that ultrasound guided CNB of STL is a safe and effective procedure that offers high diagnostic accuracy, in a dedicated surgical outpatient setting with immediate biopsy at first contact. A large fraction of all consecutive ultrasound guided CNBs in our cohort were conclusive biopsies (88.5%). In patients who underwent surgical resection after CNB, the concordance of dignity, tumor entity and histopathology grading between biopsy and surgical specimen were 97.2%, 92.7% and 92.5% respectively. This is consistent with data from a recently published meta-analysis comparing CNB with incisional biopsy reporting a 95% confidence interval (CI) ranging from 0% to 10% for non-diagnostic samples after CNB, including CT guided biopsies [15]. In this study, the accuracy for STS histotype after CNB was 88% (95% CI: 86% to 90%). Likewise, studies on ultrasound guided CNB report similar results, with rates of accurate diagnosis between 79% and 97% [17,18,19,20,21,22,23,24]. In our cohort, the cut-off value for lesion size determined by ROC analysis was 6.25 cm, lesions below this threshold were associated with a higher risk of non-diagnostic biopsy. This fits well with the concept of obtaining CNB in lesions exceeding the diameter of a golf ball (approx. 4.3 cm) [27].

The complication rate in our cohort was very low; major complications were observed in only three patients (0.8%). Looking at the details, these complications could only have been avoided potentially by CT-guided biopsy–thus requiring a much higher diagnostic effort. Again, these results are consistent with the literature where a pooled complication rate of 1% after CNB is reported [15]. Interestingly, all major complications in our cohort were observed after biopsy of lesions of the trunk or of lesions in retroperitoneal or intraabdominal locations, while no major complications were observed after biopsy of lesions of the extremities. Biopsies of lesions at the extremities may be associated with an even lower risk than at other sites. Yet, the complication rate in our cohort is too low to statistically test this hypothesis. Interestingly, no case of relevant periinterventional bleeding or hematoma was observed, including the 15.8% of patients on anticoagulant therapy. Moreover, not a single case of biopsy tract seeding was observed. CNB was conducted in an ambulatory setting in most cases (87.6%), and when hospitalized patients underwent biopsy, CNB was usually not the reason for hospital admission (63.3%). These results are in accordance with the observations of Walker et al. who reported CNB to be safe in an ambulatory setting [28].

A common alternative to ultrasound guided CNB is a CT guided approach. CT guided biopsy is associated with radiation exposure, requires additional planning, and thus bears the risk of a delay in diagnosis. In contrast, ultrasound guided CNB is less invasive, less complicated to perform, cheaper, and more readily available. An argument in favor of CT guided biopsy is that a more reliable result could be obtained in tumors where different histologic components are suspected based on imaging (e.g., a high-grade and a low-grade part of the lesion). In our study, comparison of FNCLCC grading between CNB and resection specimen showed concordant grading in 92.5% of the cases, indicating that when ultrasound guided CNB seems feasible, CT guided biopsy may not be the method of first choice. However, comparison of grading between CNB and final histopathology after surgical resection was performed only in a fraction of the patients in our study (*n* = 80), since grading was not available in all cases (e.g., pretreated resection specimen).

A recent meta-analysis claimed that ultrasound guided CNB should be performed by expert radiologists instead of surgeons. The type of CNB operator was the only variable significantly associated with heterogeneity in this study (*p* = 0.033) [29]. However, immediate ultrasound guided CNB by a qualified radiologist is often not available at first presentation of patients with STL, even in specialized centers. As a standard operating procedure at our department, ultrasound guided CNB is performed by the attending surgeon, preferably at first presentation of the patient in the outpatient clinic. This approach seeks to shorten the time between first presentation and diagnosis in order to reduce the diagnostic delay in patients with soft tissue tumors [30,31]. In our cohort, ultrasound guided CNB in the hands of surgeons resulted in high diagnostic accuracy, the latter probably depending more on the expertise of the individual operator and less on the specialty the person belongs to. To investigate this further, we analyzed the influence of the individual experience of the CNB operator on diagnostic accuracy in our cohort. Neither the affiliation of the operator to the core sarcoma team of our institution nor the number of CNBs performed had a significant impact on the rate of conclusive biopsies. However, all biopsies were performed under supervision of dedicated sarcoma surgeons, and this was not a controlled study evaluating the performance of individual CNB operators. It is reasonable to assume that ultrasound guided CNB, like any procedure, requires a certain amount of practice and has a distinct learning curve. Nevertheless, at least in the setting of a specialized center, designated expert radiologists seem not to be required to perform ultrasound guided CNB of STL to obtain good results.

This study has some limitations. This was a single center retrospective analysis, which carries an inherent risk of bias and limits the validity of the findings. There may be unknown factors that were not assessed. Furthermore, the study population was heterogeneous regarding patient history, tumor entity and lesion location; however, the respective subgroups were relatively large compared with other studies.

## 5. Conclusions

Ultrasound guided CNB of STL is a safe and effective procedure with high diagnostic accuracy. It can be performed at the first outpatient presentation and by this speed up the diagnostic and therapeutic cascade in sarcoma patients. For patients with STL larger than the diameter of a golf ball, primary ultrasound guided CNB in an outpatient setting by a qualified surgeon is recommended. CT-guided biopsy or incisional biopsy, as well as biopsy under inpatient conditions should be reserved only for cases that are not amenable to ultrasound guided CNB.

## Figures and Tables

**Figure 1 cancers-13-05659-f001:**
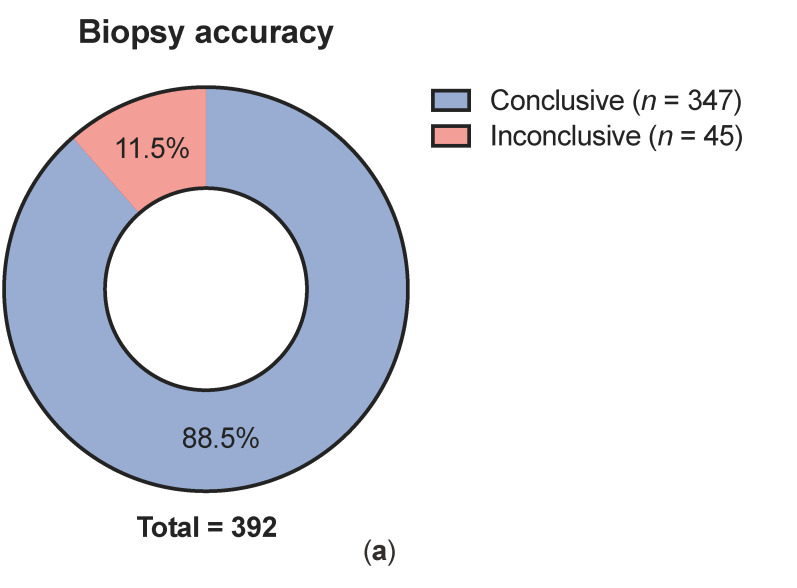
Results of ultrasound guided CNB for soft-tissue lesions. Accuracy (**a**), histology (**b**) and lesion type (malignant solid tumors (**c**)). CNB: core needle biopsy.

**Figure 2 cancers-13-05659-f002:**
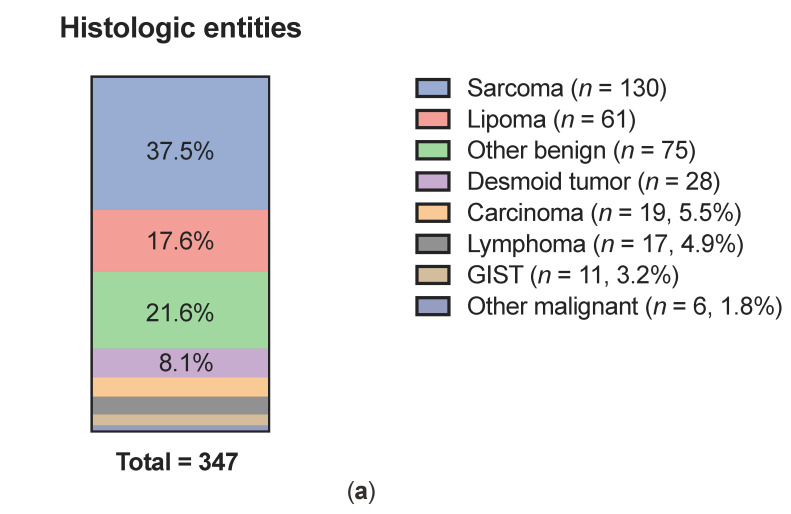
Histologic entities of CNB (**a**), sarcoma subtypes (**b**) and sarcoma gradings (**c**). GIST: gastrointestinal stromal tumor; MPNST: malignant peripheral nerve sheath tumor; NOS: not otherwise specified; SFT: solitary fibrous tumor.

**Figure 3 cancers-13-05659-f003:**
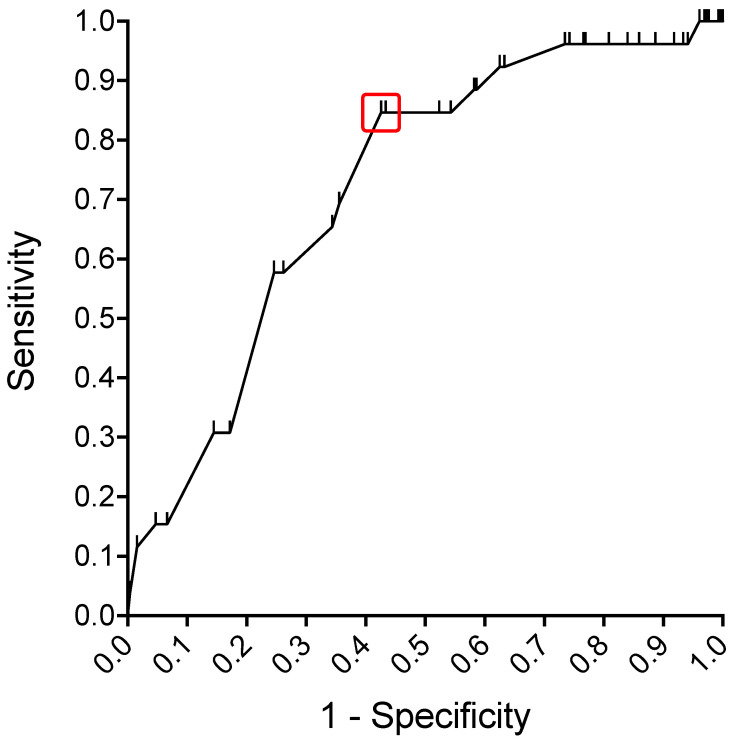
ROC curve for determination of an optimal lesion size cut off value to separate patients with conclusive CNB from patients with inconclusive CNB (red marker, sensitivity 84.6%, specificity 57.4%, Youden index 0.42).

**Table 1 cancers-13-05659-t001:** Clinical characteristics of CNB patients.

Variable	n or Median	% or IQR
**Gender**		
Female	201	51.3
Male	191	48.7
**Age (years)**	59	46–71
**History of tumor**		
Total	93	23.7
Sarcoma	40	43.0
GIST	9	9.7
Desmoid tumor	14	15.1
Colorectal cancer	7	7.5
Lymphoma	5	5.4
Other	18	19.4
**Time to biopsy (months)**	2	1–5
**Lesion size (cm)**	8	5–12
**Location of lesion**		
Lower extremity	137	34.9
Upper extremity	32	8.2
Intraabdominal or retroperitoneal	77	19.6
Trunk	146	37.2
**Biopsy setting**		
Outpatient	343	87.5
Inpatient (for biopsy)	18	4.6
Inpatient (other reasons)	31	7.9
**Anticoagulation**		
Total	48	15.8
Acetylsalicylic acid	37	77.1
Direct oral anticoagulants	9	18.8
Other	3	6.3

CNB: core needle biopsy; IQR: interquartile range.

**Table 2 cancers-13-05659-t002:** Factors associated with inconclusive CNB.

Variable	*p* Value			
	Lower Extremity	Trunk	Upper Extremity	Intraabdominal/Retroperitoneal
Lesion size	0.036 *	0.006 *	0.819	0.537
Physician experience (<10 CNB vs. ≥10 CNB)	1.000	0.435	0.159	0.440
Physician STT team affiliation (yes vs. no)	1.000	0.625	0.560	0.722
Time to biopsy	0.913	0.838	0.303	0.143
Biopsy setting (outpatient vs. inpatient)	1.000	0.670	0.431	0.531
Anticoagulation (yes vs. no)	0.562	0.207	1.000	0.681
History of malignancy (yes vs. no)	0.607	1.000	0.254	0.750
Gender (female vs. male)	0.682	0.812	0.079	0.396
Age	0.088	0.063	0.171	0.244
Complications (yes vs. no)	—	1.000	—	1.000

CNB: core needle biopsy; STT: soft tissue tumor. Asterisks (*) indicate statistical significance.

## Data Availability

The data presented in this study are available upon reasonable request from the corresponding author. The data are not publicly available due to ethical restrictions and data protection regulations.

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
