# Peer review of "Accuracy and Safety of Ultrasound-Guided Core Needle Biopsy of Soft Tissue Tumors in an Outpatient Setting: A Sarcoma Center Analysis of 392 Consecutive Patients"

_cancers, 2021, doi:10.3390/cancers13225659_

Round 1
Reviewer 1 Report
Simple Summary: line 22 authors need to define what is meant by concordance with "dignity" and "tumor entity"; line 24 insert "in" after occurred; line 25 authors need to define in layman's terms what is meant by "unselected patients"
Abstract: It summarizes key findings. It is concise and contains necessary information. As above, authors need to define what is meant by concordance with "dignity" and "tumor entity" (line 34)
Introduction: line 53 change "regarding complications" to "in regards to complications"; line 59 remove "attitude" and replace with "ability". Please remove author instructions line 60 thru line 68
Materials and Methods: line 75 remove "(sarcoma)"; line 86 spell out acronym ASS and describe anticoagulation used (move information from lines 168-169 to here); line 90 change "in" to "under" and change "under" to "using". Overall methodology described is appropriate and data is collected appropriately.
Results: lines 137-140, authors should clarify the difference between confirmed malignancy versus patients with history of malignancy in association with older age. It is confusing. Section outpatient and inpatient setting: title, abstract and introduction focuses perspective of study on outpatient setting for CNB. Why is inpatient data included in study? Figure A3 (awkward). Authors should re-order figures 1 through 4; line 187 authors should clarify that classified as high grade includes FNCLCC grades II and III. Line 200 change Table 2 to Table 1. Lines 201-206 authors explain size as only significant factor associated with an inconclusive biopsy, but authors should review pathology (histology slides) and mention if necrosis and/or hemorrhage was considered as factors (sampling error) and what about interpretation error as a factor? Lines 213-214 as previously mentioned, authors need to clarify what is meant by "tumor dignity" and "tumor entity" Lines 219-221 remove (author instructions/guidelines)
Discussion: line 237 remove "a" after offers; lines 278-279 authors should review and re-word comment on grading between CNB and final. Pathologists make every effort to grade pretreatment biopsy (precluding factors usually are necrosis or insufficient tumor to evaluate appropriate # fields for mitotic count if using FNCLCC grading). Post treatment tumor is not graded and is recommended not to be graded because more often than not, it is not reflective of true grade and high grade tumors are inadvertently graded as low grade. Immunostains help determine line of differentiation and if tissue insufficient also a factor precluding grading; line 291 remove "she" and insert "the person"; line 300 remove "with good results" and replace with "to obtain good results." Authors should expand upon limitation factors providing specific factors considered
Conclusions: line 307 remove "in all common locations"; authors should clarify the use of the term "unselected" patients; line 31 authors make a claim that ultrasound guided CNB should be performed by treating surgeon, why? Authors indicated that study did not delve into evaluation of CNB operator. It can't be a trainee under the supervision of surgeon, nurse practitioner, etc.
Figure 2: Please review to make consistent with (n=number, %) in all parts
Table 1: incorrectly labelled table 2
Reviewer 2 Report
In general:
The manuscript describes a study of evaluation of accuracy and safety of ultrasound-guided core needle biopsy. The monocentric study includes 392 patients with unclear soft-tissue tumors of trunk and extremities. The evaluation of the accuracy was done by the comparison of histology findings between the biopsy and resected tumor. The cohort of patients is very heterogeneous. The results show the high grade of accuracy of the ultrasound-guided core needle biopsy and no relevant number of major complication during/after the biopsy (high safety).
Major issues:
Materials and Methods: I am missing the definition of entities according WHO-classification und grading according FNCLCC.
Materials and Methods: The definition of accuracy is absent. Please clarify your definition. The accuracy of biopsy should be correlated with histologic finding of resected tumor. However, only 179 cases underwent the tumor resection (less than 50%). What is about the accuracy of 213 cases without resection?
Line 184-185: Evaluation of the grading accuracy: there are no information of the neoadjuvant therapy in sarcoma cases. In the case of neoadjuvant therapy, the confirmation of the grading it the tissue of resected tumors is not allowed. What was the procedure of the evaluation of accuracy of grading determination?
Discussion: is the comparison of the cases with primary and recurrent tumors sensible? Are these groups comparable? Please discuss it.
Discussion Lines 226-235: Then what is the advantage of ultrasonic control? Please discuss in more detail.
Minor issues:
Simple summary (Lines 17 and 26): in these lines, authors mention the word “sarcoma”. However, the cohort includes a big number of benign lesions. “Sarcoma” should be replaced by “soft tissue leasion”.
Line 45: “In soft tissue lesions (STL), treatment planning largely depends on histology.”. Please replace “histology” by the “entity”.
Line 50: “…(CNB) is recommended…“ You have to describe the incisional biopsy as alternative procedure. CNB is not without alternative, at least in extremities.
Lines 19 and 58: “… and histologic results…”. The evaluation of accuracy means the evaluation of the histologic findings. It is a repletion, please delate “histologic results”.
Lines 60-68: I do not think, these lines are a part of manuscript. Please delate.
Lines 75-77: This information contains no relevant information for the study. Please delate it.
Line 124: please, note the absolute number of cases with known malignant diagnosis
Lines 124-126: I cannot understand this sentence: Did these patients underwent the primary biopsy outside of sarcoma center and the biopsy was repeated? Or had these patient a recurrence disease? Please clarify.
Table 1: History of malignancy, desmoid tumor: desmoid tumor is not malignant. Please rename the line with ”History of malignancy” to e.g. “Recurrent tumor”
Figure 1c: the number of malignant tumors differs to the number of malignant tumors in 2b. Please correct or clarify it.
Figure 2a: there 17 lymphomas. Which location had these tumors? Were these the biopsies of lymph nodes? If it is correctly, please note this information in “Materials and Methods”
Lines 219-221: Again, I do not think, these lines are a part of manuscript. Please delate.
Lines 238-241: I cannot comprehend this data based on results of the manuscript.
Lines 269-270: What makes the CT-guided biopsy less invasive compared to the US-guided biopsy?
Lines 278: the number of determinated grading cases (n=80) contradicts the number in figure 2c (n=107). Please clarify.
Reviewer 3 Report
This paper is a retrospective analysis of feasibility and accuracy of core needle biopsy ( CNB ) for soft tissue lesions. The idea of core needle biopsy is not entirely new but this paper described the experience of a relatively large series
It would be helpful if the authors can clarify the following issues
- Please state clearly how many of the patients who underwent CNB eventually underwent surgical excision to confirm the histology. Specifically, please confirm if surgical excisions were done for lesions whose CNB showed benign histology
- The practice of CNB of the intra-abdominal / retroperitoneal lesions is more controversial. One of the concerns is the possible seedling of malignant cells into the peritoneal cavity or rupture of a lesion which was not entirely solid. Do the authors have any data on the patient's operative findings/ prognosis etc ?
- The authors suggested that surgeons instead of radiologists should perform the CNB. Please kindly clarify if the surgeons had undergone ultrasound training to accurately identify anatomical structures , especially intra-abdominal structures ?
Round 2
Reviewer 3 Report
Thank you for your revision. I have no further comments.